# BIAS LEARNING: QUANTIFYING AND MITIGATING POSITION SENSITIVITY IN TEXT EMBEDDINGS

## ABSTRACT

Embedding models are crucial for tasks in Information Retrieval (IR) and semantic similarity measurement, yet their handling of longer texts and associated positional biases remains underexplored. In this study, we investigate the impact of content position and input size on text embeddings. Our experiments reveal that embedding models, particularly APE- and RoPE-based models, disproportionately prioritize the initial portion of the input. Ablation studies demonstrate that insertion of irrelevant text or removal at the start of a document reduces cosine similarity between altered and original embeddings by up to 12.3% more than ablations at the end. Regression analysis further confirms this bias, with sentence importance declining as position moves further from the start, even with with content-agnosticity. We hypothesize that this effect arises from pre-processing strategies and chosen positional encoding techniques. To address this, we introduce a novel data augmentation scheme called Position-Aware Data Sampling (PADS), which mitigates positional bias and improves embedding robustness across varying input lengths. These findings quantify the sensitivity of retrieval systems and suggest a new lens towards long-context embedding models.

## 1 INTRODUCTION

Embedding models are increasingly used to encode text in critical applications like document search systems. Along with the rise of long-context models, there has been growing research on model performance based on input position (Nelson F. Liu, 2023), but current work remains limited to encoder-decoder and decoder-only models. Embedding models, in contrast, are theoretically position-invariant, due to their lack of the causal attention mask.

In this study, we investigate the influence of content position and input size on the resulting text embedding vector from eight embedding models. Our findings reveal a systematic bias in which embedding models, disproportionately weigh the beginning of a text input. This results in greater importance being assigned to the initial sentences of multi-sentence or long-context inputs. To demonstrate this, we conducted two types of ablation studies: one involving the insertion of irrelevant text ("needles") at different positions in the document (Guerreiro et al., 2023), and another involving the removal of varying text chunks. We observe that, dependent on positional encoding mechanism, inserting irrelevant text at the beginning of a document reduces the cosine similarity between the altered and original document embeddings by up to 8.5% more than when inserted in the middle, and 12.3% more than when inserted at the end. Similarly, removal experiments show that the largest decreases in similarity occur when text is removed from the beginning of the document.

To further explore this bias, we employ regression analysis to measure sentence-level importance on a complete document-level embedding, isolating model position bias from human writing patterns. Our analysis shows a significant decline in regression coefficients as the sentence position moves further from the beginning of the document, reinforcing the bias toward earlier content. To rule out dataset-specific effects, we repeat all experiments with randomly shuffled sentences and obtain similar results, confirming that this bias arises from the model's internal mechanisms rather than document structure.

We hypothesize that this bias stems from common pre-processing strategies used during training when the input exceeds the model's context window (Liu et al., 2019; Xiao et al., 2023). This has important implications for real-world retrieval tasks, where documents with key information located

later in the text may be overlooked due to the model's disproportionate weighting of early content (Barnett et al., 2024).

We conclude by discussing the broader implications of these biases in embedding models and highlight the need for future research to develop methods that can better handle the entirety of long-context inputs without disproportionately prioritizing the beginning.

## 2 BACKGROUND

### 2.1 NOISE FROM DOCUMENT CHUNKING FOR IR TASKS

In practical applications, documents often exceed the context length capabilities of embedding models, necessitating chunking strategies like naive, recursive, or semantic chunking (Fei et al., 2023; Gao et al., 2024). This process divides a document into smaller pieces that fit within a model's context window, then embeds each chunk separately for insertion into a vector database (Johnson et al., 2017) and downstream use in Retrieval-Augmented Generation (RAG) (Lewis et al., 2021) tasks. This causes an unintentional, outsized amount of noise in the beginning and end of documents as a function of selected chunking strategies. There is growing applied research in improving chunking strategies, or model inputs, to reduce the amount of noise(Unstructured, 2024; Brandon Smith, 2024). However, there is little known about what causes retrieval performance degradation on the model side.

Academic research has provided initial research into model behavior through the context window, but are primar. Previous work have studied model performance y focused on encoder-decoder and decoder-only models (Nelson F. Liu, 2023). These models incorporate a causal attention mask, which can contribute to positional bias—an overemphasis on earlier input positions—by restricting attention to past tokens during sequence generation. However, this mechanism does not account for positional bias in encoder-only models, where bi-directional attention allows the model to attend to all tokens in the sequence simultaneously.

### 2.2 BIDIRECTIONAL ENCODING IN EMBEDDING MODELS

Embedding models, particularly those utilizing transformer encoder architectures (Vaswani et al., 2023), employ layers of bidirectional self-attention blocks to process text (Devlin et al., 2019). These models are distinct from decoders in that they generate a fixed-length vector representing the entire input text. This is achieved by producing an output matrix $L \times D$ (where $L$ is the sequence length and $D$ is the dimensionality of the embeddings), and then applying either mean or max pooling across the $L$ dimension (Reimers & Gurevych, 2019). Such pooling operations are position-invariant, theoretically suggesting an unbiased treatment of input positions in terms of attention and representation (Su et al., 2023). Additionally, unlike generative models that use a causal attention mask to zero out certain elements in the softmax operation during attention calculation, embedding models are fully bi-directional and do not require an attention mask.

We use cosine similarity to compare the output embeddings from these models, especially to study the effects of textual modifications such as insertions or deletions. Cosine similarity measures the cosine of the angle between two vectors, thus providing a scale- and orientation-invariant metric to assess the similarity between two text representations (Li & Li, 2024).

Due to the invariance of the architecture and similarity measurement we employ, the last systematic source of bias stems from learned positional embeddings used in our models and the models' training methodology, which are heavily connected.

### 2.3 POSITIONAL ENCODING TECHNIQUES

**Absolute Positional Embedding (APE)** assigns fixed position-specific vectors based off of position id to each token embedding. This was first popularized by BERT (Devlin et al., 2019) and remains the most common technique to add positional information in encoder-style models today.

**Rotary Positional Embedding (RoPE)**: RoPE encodes positions by applying a rotation to each token's embedding in the 2D subspaces of the embedding space. For each embedding vector $x$, it applies a rotation matrix $R(\theta)$ based on the position $pos$:

$$\mathbf{x}_{\text{pos}}^{(2i)} = \mathbf{x}^{(2i)} \cos(\theta_{\text{pos}}) - \mathbf{x}^{(2i+1)} \sin(\theta_{\text{pos}})$$

$$\mathbf{x}_{\text{pos}}^{(2i+1)} = \mathbf{x}^{(2i)} \sin(\theta_{\text{pos}}) + \mathbf{x}^{(2i+1)} \cos(\theta_{\text{pos}})$$

where $\theta_{\text{pos}} = \frac{\text{pos}}{10000^{2i/d}}$, $i$ indexes the embedding dimensions, and $d$ is the dimensionality.

**Attention with Linear Biases (ALiBi)**: ALiBi introduces a relative bias into the attention scores rather than modifying the embeddings. The bias is linear with respect to the distance between tokens. The attention score $A(i, j)$ between token $i$ and token $j$ is modified by adding a bias term $m(|i-j|)$, where $|i - j|$ is the distance between tokens:

$$A(i, j) = \frac{\mathbf{q}_i \cdot \mathbf{k}_j}{\sqrt{d_k}} + m(|i - j|)$$

where $m(|i - j|)$ is a linear function of the relative distance between tokens $i$ and $j$, and $d_k$ is the dimensionality of the key vectors.

## 3 EFFECT OF SENTENCE-LEVEL POSITIONING IN EMBEDDING OUTPUT

We explore how the position and size of a sentence in a text influence a document's final embedding vector. Our methodology adapts the needle-in-a-haystack test (Guerreiro et al., 2023), traditionally used for generative models in information retrieval (Team et al., 2024), to evaluate embedding models.

### 3.1 EXPERIMENTAL SETUP

#### 3.1.1 INSERTION OF IRRELEVANT TEXT

We investigate the impact of adding irrelevant or adversarial text ("needle") to a document. After inserting the needle, we generate a new embedding for the altered text and compare it to the original using cosine similarity. We vary the needle's length (5%, 10%, 25%, 50%, and 100% of the original text's token count) and position (beginning, middle, end) across 15 experimental conditions. We use an extended version of Lorem Ipsum placeholder text (Timmer et al., 2022) that exceeds the length of our longest datapoint and is structured in paragraph format to achieve a needle with structural similarity to our data while avoiding a confounding effect on the embedding model.

#### 3.1.2 REMOVAL OF TEXT

In a parallel experiment, we remove portions of text (10%, 25%, 50% of sentences, rounded up) from different positions (beginning, middle, end) in the document. The resulting text is then embedded, and its similarity to the original embedding is measured using cosine similarity.

### 3.2 MODELS

We test various models, segmented by their positional encodings, to demonstrate the consistency of our results across multiple popular embedding models. We used six open-source models utilizing various positional encoding methods (Table 1). We additionally test Cohere's Embed-English-v3.0 (Reimers, 2023) and OpenAI's Text-Embedding-3-Small (OpenAI, 2024) due to their popularity and real-world applicability. Although we picked these models due to their varying positional encoding methods and performance, we acknowledge these may not generalize to other architectures and datasets. For texts exceeding these limits, we truncate from the end to fit the models' context windows.

Table 1: Models positional encodings and context window size

| Positional Encoding | Model | Context Size |
|---|---|---|
| APE | BGE-m3 (Chen et al., 2024) | 8912 |
| | E5-Large-V2 (Wang et al., 2022) | 512 |
| RoPE | Nomic-Embed-Text-v1.5 (Nussbaum et al., 2024) | 8192 |
| | E5-RoPE-base (Zhu et al., 2024) | 512 |
| ALiBi | Jina-Embeddings-v2-Base (Günther et al., 2024) | 8192 |
| | Mosaic-Bert-Base (Press et al., 2022) | 1024 |
| Unknown/Closed-Source | Text-Embedding-3-Small (OpenAI, 2024) | 8191 |
| | Embed-English-v3.0 (Reimers, 2023) | 512 |

### 3.3 DATASETS

To minimize dataset bias and validate our findings across diverse text types, we selected and used 200 examples each from the following datasets to represent a range of writing categorizations and lengths: **PubMed Publications**, We use PubMed publication abstracts Cohan et al. (2018) to assess the impact of our ablations on scientific writing. Scientific texts are characterized by their structured presentation of information and specialized vocabulary. Understanding how embeddings capture this complexity can provide insights into their utility in academic and research applications; **Paul Graham Essay Collection**, We analyze over 200 essays written by Paul Graham Goel (2024), varying from 400 to 70,000 words. Paul Graham's essays are known for their thoughtful, reflective style and coherent argument structure, making them ideal for studying how embeddings handle nuanced and complex idea development over long texts; **Amazon Reviews**, Drawn from MTEB's Amazon Polarity dataset Zhang et al. (2016), this helps us examine consumer review text. Reviews are direct and opinion-rich, offering a perspective on how embeddings process everyday language and sentiment, which is crucial for applications in consumer analytics; **Argumentative Analysis**, From the BiER benchmark's Argumentative Analysis (ArguAna) dataset Wachsmuth et al. (2018), we explore embeddings of formal persuasive writing. This dataset includes well-constructed arguments that are ideal for testing how embeddings capture logical structure and the effectiveness of rhetoric; **Reddit Posts**, More Informal and diverse writing styles can be found on Reddit Geigle et al. (2021). This dataset introduces grammar, style, and subject matter diversity into our tests, extending our findings to be more robust and adaptable to a wide range of writing styles.

### 3.4 RESULTS AND DISCUSSION

Our results indicate a pronounced drop in similarity when irrelevant text is inserted at the beginning of documents, with less impact observed when additions occur in the middle or end. Specifically, for APE models, introducing an insertion equal to 20% of the total content at the beginning results in an average cosine similarity of 0.885, compared to 0.963 at the end—a relative decrease of approximately 8%. RoPE-based models show a stronger sensitivity to this disruption, with cosine similarity dropping to 0.819 at the beginning, a 15.4% decrease compared to the 0.968 similarity at the end. By contrast, ALiBi models are the most robust, maintaining a high cosine similarity of 0.981 at the beginning and 0.999 at the end, reflecting only a 1.8% decrease (Figure 1).

This suggests that earlier positions in the input sequence play a more critical role in model performance, and different positional encoding methods, in particular those that require learned parameters (APE and RoPE), are less robust to this type of input perturbation.

This trend persists across all insertion sizes, with larger insertions intensifying the drop in similarity. Even though the magnitude of the degradation varies by model, we find the trend robust to model differences . Across all five models tested, the average decrease in cosine similarity is approximately

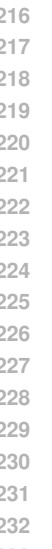
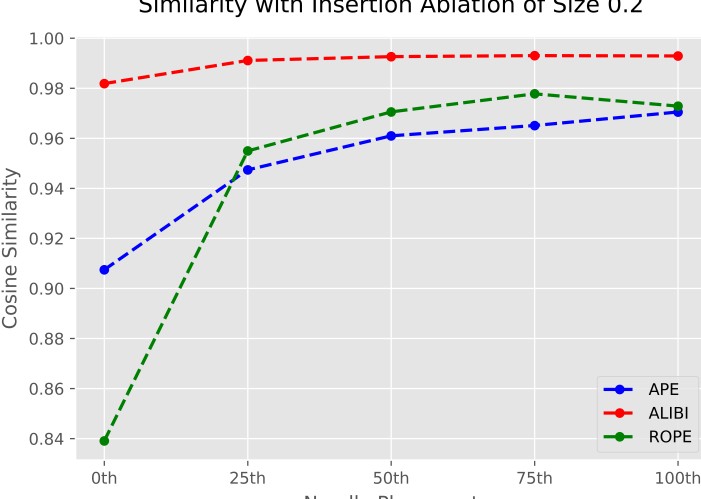

Figure 1: Cosine similarity vs. insertion needle position. The needle is comprised of irrelevant text that is 20% of document size.

7%, indicating a consistent pattern of sensitivity to input alterations at the beginning of the sequence (Appendix A).

Notably, even significant alterations where half of the text is irrelevant still retain a minimum similarity of 0.7, suggesting an unexpected robustness of the embeddings to extensive modifications. We leave investigation of this behavior to future work.

Table 2: Cosine similarity vs. needle position, averaged across all ablation sizes as a percentage

|  | **Positional Encoding** | 0th | 50th | 100th |
|---|---|---|---|---|
| Insertion | APE | 88.53 | 95.1 | 96.27 |
|  | RoPE | 81.89 | 96.43 | 96.82 |
|  | ALiBi | 97.95 | 99.13 | 99.21 |
| Removal | APE | 92.86 | 96.67 | 97.22 |
|  | RoPE | 87.61 | 97.33 | 97.43 |
|  | ALiBi | 99.4 | 99.89 | 99.91 |

Additionally, we observe that removal ablations yield similar results, although the overall similarity scores are higher in comparison to insertion ablations (Table 2). Removing half of the sentences from the beginning results in a median similarity that is 10.6% lower than when sentences are removed from the end, with no significant difference between middle and end removals. Interestingly, even a 50% text removal from the middle maintains a median similarity of 95%, corroborating our findings from the insertion experiments, where a large drop in similarity was expected but not observed (Appendix B). These results suggest that the removal of content has similar impacts to the insertion of irrelevant text, albeit introducing less noise to overall similarity scores.

## 4 ANALYSIS OF EMBEDDING DECOMPOSITION

Recent advancements in embedding interpretability have demonstrated that certain dimensions in high-dimensional semantic spaces may correspond to specific linguistic or semantic features, such as sentiment or subject matter (Dar et al., 2023). Further research has shown that vector operations,

such as adding embeddings, can produce new vectors that represent the semantic meaning of their components (Senel et al., 2018).

Building from these works, we explore the impact of sentence-level positioning on the final document embedding vector through regression analysis, which offers a more direct method to quantify the contribution of individual sentences to a document's embedding representation.

Human writing often emphasizes key information at the beginning and end of documents, a technique that may introduce biases in datasets and reason for embeddings to skew towards these positions. To address these, we employ additional data augmentation and ablation techniques aimed at isolating and understanding these effects, to ensure that our findings more accurately reflect model behavior rather than dataset peculiarities.

## 4.1 RECONSTRUCTING EMBEDDING VECTORS THROUGH LINEAR COMBINATIONS OF CONSTITUENTS

To start, we wanted to validate the assumption that the sentence embeddings of a larger document can meaningfully be used as a proxy for the original document embedding (Tsukagoshi et al., 2022).

To test this, we wanted to determine how much reconstruction loss we would incur from using an optimal linear combination of sentence embedding vectors instead of a full multi-sentence embedding vector. Optimizing for train $R^2$, we use Ordinary Least Squares (OLS) regression to reconstruct the document embedding from its sentence embeddings, with the multi-sentence embedding vector as our response and each sentence vector as a predictive datapoint for our regression. Our model choice is notable for its direct interpretability (Słoczyński, 2020), though we acknowledge and check for potential issues posed by OLS, such as multicollinearity. Our regressions use normalized embeddings (L2 norm of 1) to ensure scale invariance (Steck et al., 2024). We separate our data points into their component sentences by use of punctuation such as periods, and new lines.

When we regress the sentence embedding vectors onto the multi-sentence embedding vector, we find that our train $R^2$ across the eight models and five datasets we used ranges from 0.75 to 0.99, with an average $R^2$ or 0.876 when reconstructing the multi-sentence embedding vector. This result indicates that approximately 87.6% of the variance in a long-content document embedding can be accounted for by analyzing the embeddings of the individual sentences constituting the document. The Mean Squared Error (MAE) summed over all dimensions of this reconstruction across all models and datasets ranged from 0.001 and 0.01 with an average of 0.0069, suggesting minimal deviation in the reconstructed vectors (Appendix D.

## 4.2 ANALYZING REGRESSION COEFFICIENTS AS IMPORTANCE WEIGHTS

Given the high explanatory power of our regression models, the coefficients given to each sentence (datapoint) in our regression are strong indicators to determine their relative importance to the total document. To standardize our comparisons across documents, we standardized each coefficient vector by its L2 norm. One potential issue to note with this approach is the presence of negative coefficient values, but these tended to be rare and very low in magnitude, with very little influence on our final analysis.

We judge the importance of a sentence by its regression coefficient. For example, if a regression on a two-sentence document yielded weights 0.8 and 0.6, we conclude that the first sentence is 33.3% more important to the final semantic meaning of the text than the second sentence.

There is a downward trend in coefficient values with increasing sentence position, suggesting a positional bias where earlier sentences generally have a greater impact on the document's overall semantic representation. To quantify this observation, we plot regression coefficients against sentence positions over all the documents in our dataset (Figure 2).

## 4.3 EMBEDDING POSITIONAL BIAS IS ROBUST TO HUMAN-LEVEL WRITING BIAS

To validate that this observed bias is not solely a byproduct of dataset-specific characteristics, namely human-level writing bias, we conducted additional regression experiments where all sentences from the above pre-processing steps were shuffled before their embeddings were generated. Using these

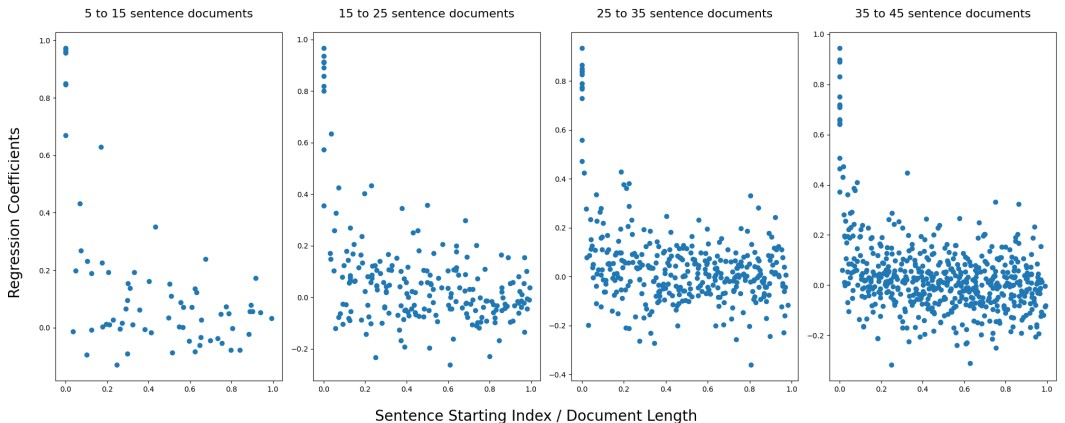

Figure 2: Regression coefficients vs. sentence position, bucketed by document length.

Table 3: Correlation and statistical significance of sentence position against shuffled text

| Positional Encoding | Correlation | P-value |
| --- | --- | --- |
| APE | -0.127657 | 2.233374e-103 |
| RoPE | -0.115861 | 2.259581e-85 |
| ALiBi | -0.07615 | 9.205763e-38 |

new embeddings, remarkably, the results mirrored the original findings, with the randomly selected first sentence in the shuffled document consistently receiving a higher weight, thereby disambiguating our results from potential dataset biases.

More specifically, we expect the weight assigned to the first sentence to follow a uniform weight of $\frac{1}{num\_sentences}$. However, this analysis shows a distinct negative correlation between sentence position and importance score, with significant deviations from the expected uniform distribution ($\alpha \ll 0.001$), confirming a systematic positional influence within document embeddings as shown in Table 3. These findings suggest that the embedding models may inherently prioritize the initial information presented in any text sequence, irrespective of its original position in the document. Further results, broken down by sentence length, can be found in Appendix C.

## 5 ISOLATING THE ROLE OF TRAINING METHODOLOGY IN MODEL BIASES

During training, input data is processed sequentially, starting at the beginning of the context window. Variable-length training samples are packed into this fixed window, often necessitating truncation when the input exceeds the window's length. Truncation typically discards content from the end, leading to a systematic bias where earlier positions in the sample receive disproportionate attention. As shown in the previous experiments, this systematic truncation is not merely a technical necessity but a fundamental design choice that influences model behavior, as the initial sections of documents - typically containing abstracts or executive summaries - are disproportionately represented.

For a given position $i \in [0, N]$ within a context window of length $N$, the model observes $t_i$, the number of non-padding tokens encountered at position $i$. The importance of position $i$ can then be modeled as $imp(t_i) = u(t_i)$, where $u(\cdot)$ represents the model's updates based on the presence of non-padding tokens at $t_i$.

As traditional truncation favors earlier positions, the frequency with which tokens are seen at the beginning of the context window is inherently higher than at the end. This can be modeled as a

monotonically decreasing function, where the quantity of non-padding tokens at $t_i$ diminishes as $i$ increases. As a result, the relative importance of earlier positions $imp(t_1) \geq imp(t_2) \geq \cdots \geq imp(t_N)$ is systematically higher, introducing an implicit bias that prioritizes early context over later content.

Although this monotonic impact on position can theoretically be removed by maintaining an equal number of effective updates throughout the context, it is unknown what the impacts on computational costs, and model performance would be. Future pre-training, as well as employing novel context-length enhancement methods, with this bias in mind will require additional research to fully understand the impacts, leading us to believe that this bias will continue in future models.

### 5.1 Is it possible to remove positional bias in post-training?

Following our theory on bias learned through the pre-training process, we experiment with smaller, cost-effective fine-tuning methods to remove this bias. We do this by fine-tune models to use data without the front-truncation, yet still holds similar semantic meaning to the initial data points.

We propose a new framework, Position-Aware Data Sampling (PADS), where subsets of data points are randomly sampled based on input position, to solve this positional bias. The method augments the data by inputting training points that would normally be truncated, and randomly selecting subsets of each data point based on position away from the beginning of the original input. For example, instead of front-truncating 50% the length of a given example, we select uniformly a token position from 0 to n/2, where n is the token length of the data point.

In our fine-tuning experiments, we create positive pairs by sampling from each original twice. For negative pairs, we sample once from both the original and another random data point in the dataset. Using these pairs, we use contrastive loss to fine-tune the model towards our goal. We follow these steps for three datasets and using this to fine-tune BAAI's BGE-small-en-v1.5. The three datasets included are the Paul Graham Essay Collection, PubMed Publications, and Amazon Reviews. We sample a maximum of 20% from each dataset, selecting 50 examples for the Paul Graham dataset and 225 for the other two datasets. Following the procedure above, we select 50%of each original datapoint and create a positive and negative pair from each, resulting in an augmented dataset of 1000 examples. We use cosine similarity within our contrastive loss function, and then use this with the Adam optimizer for three epochs.

Table 4: Average cosine similarity between original and ablated inputs

| Model | Beginning | Middle | End |
|---|---|---|---|
| Original | 0.923 | 0.979 | 0.983 |
| Finetuned | 0.984 | 0.993 | 0.993 |
| Percent Improvement | **6.1%** | **1.4%** | **1.0%** |
| | | | |
| Original (external datasets) | 0.920 | 0.978 | 0.982 |
| Finetuned (external datasets) | 0.988 | 0.995 | 0.995 |
| Percent Improvement | **6.8%** | **1.7%** | **1.3%** |

With this new method, we have been able to effectively remove positional bias and improve similarity metrics to levels similar to when ablations are put in positions different from the beginning. The new model has been able to reduce bias by 6.9% with insertion needles, and 6.1% averaged between insertion and removal ablations. This work suggests that models can learn to fix its early positional bias by sampling the subset position of the input it is training on, and is notable for its simplicity in implementation.

## 6 Limitations

We have limited our claims to using 6 models with 6 datasets, but this can be extended to look at positional bias for more models and datasets, particularly those outside of English, to eliminate

implicit bias from the experimental design. Additionally, the fine-tuning method can be adopted to the pre-training method to look at the full effects and performance impacts, outside the post-training context.

## 7 FUTURE WORK

Future work incorporating our findings can focus on three distinct directions:

**Alternative Evaluation Metrics**   Exploring alternative evaluation metrics beyond cosine similarity is essential to assess the effectiveness of embedding models. Future research should consider metrics such as Word Mover's Distance (WMD) Kusner et al. (2015) for capturing semantic similarity, BERTScore Zhang et al. (2020) for evaluating contextual alignment, and NDCG (Normalized Discounted Cumulative Gain) Wang et al. (2013) for ranking quality in information retrieval tasks. Additionally, task-specific metrics like classification F1-score, BLEU Papineni et al. (2002) for translation quality, and ROUGE Lin (2004) for summarization accuracy can provide deeper insights into model performance.

**Model Architecture and Training Process Innovations**   Given our findings, model creators can employ alternative training techniques such as sentence shuffling or random truncation of long texts during the embedding training process. These methods can help mitigate positional biases and enhance model robustness, both in the pre- and post-training phases. Since embedding models use contrastive loss Mnih & Teh (2012) rather than classification loss like generative models, careful consideration is needed to determine the best way to compare these ablations with their original texts. This could involve designing new contrastive learning objectives that account for the positional integrity of the input text. Additionally, incorporating architectural modifications, such as advanced attention mechanisms or positional encodings Press et al. (2022), can further reduce biases and improve the models' ability to handle long-context inputs.

**Improved Document Chunking and Impact on Downstream Information Retrieval Tasks**   Future work should focus on how this analysis may advise future chunking techniques. By aligning chunking strategies with the positional bias, we can create more effective strategies, for example, having helpful context in the front of each chunk as opposed to having potential noise from the chunking split. Evaluating various existing chunking strategies in existing literature can reveal how different approaches affect the retrieval accuracy and relevance of results. This integrated approach would provide a more performant system for downstream retrieavl tasks.

## 8 CONCLUSION

Our study uncovers a positional bias in embedding models, where sentences at the beginning of a document disproportionately influence the resulting embeddings. This bias is consistent based on the positional encoding technique within each observed models with different context sizes and datasets and is evident in both text insertion and removal experiments. We further study this effect by analyzing the effects of individual sentence on the total embedding to removing human writing bias from the dataset, isolating the effect of model positional bias. We find that models, despite their positional encoding, exhibit this model preference for earlier content. We continue this by offering an explanatory framework around training methodologies as the proposed cause of the bias. Finally, we explore a potential sampling techniques during post-training to mitigate this bias.

Positional bias presents significant challenges in critical applications like information retrieval in document search systems, where suboptimal chunking or poorly structured documents can disproportionately degrade retrieval performance. Furthermore, as research into extending context length advances—particularly with continued training on longer sequences—there is growing evidence that this phenomenon warrants deeper exploration and innovative solutions.

These insights underscore the need for revised training strategies that address positional biases to produce more balanced semantic representations. While our initial experiments demonstrate the potential of fine-tuning to reduce this bias, additional research is crucial to develop robust techniques

that fully mitigate positional biases. By refining training methodologies, we can achieve more consistent and unbiased model performance across various tasks and contexts.

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

## A   COSINE SIMILARITIES ACROSS INSERTION ABLATION SIZES AND DATASETS

The following are the results of running insertion and removal ablations of given sizes on input examples. These are the results of the average cosine similarity across all datasets.

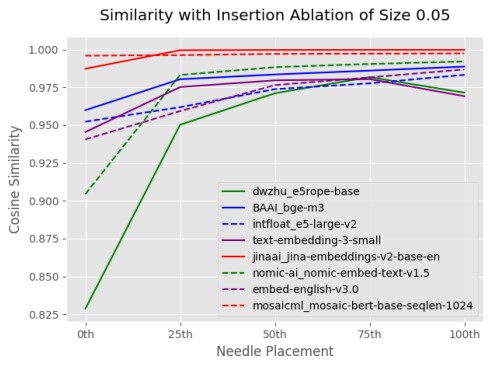
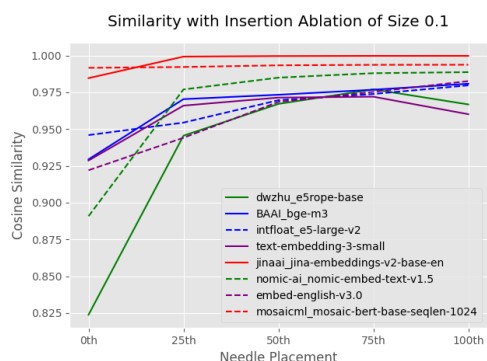

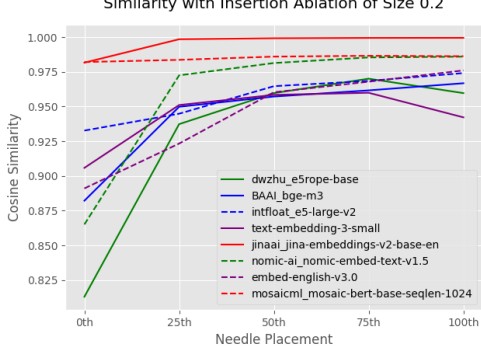
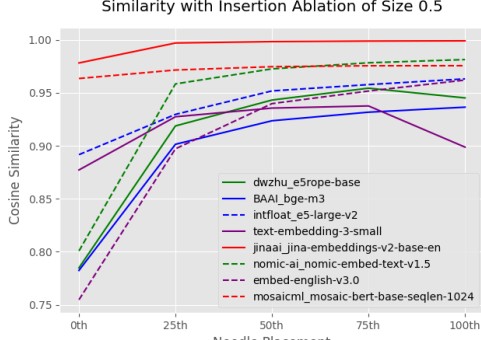

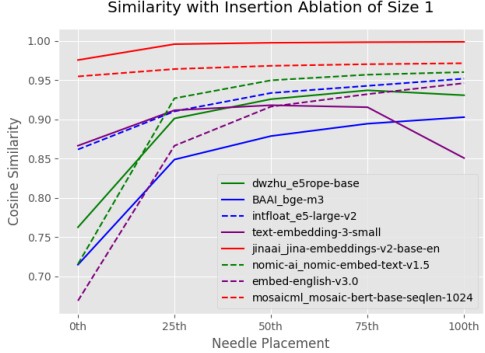

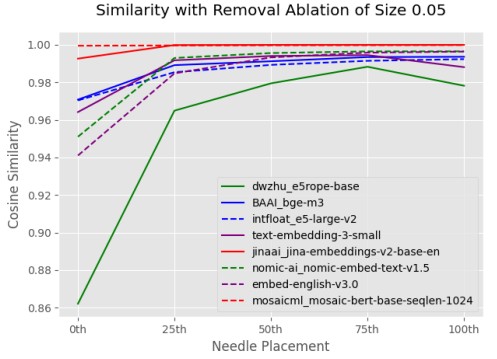

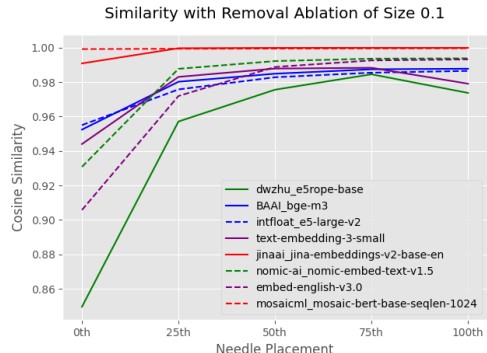

## B COSINE SIMILARITIES ACROSS REMOVAL ABLATION SIZES AND DATASETS

## C SENTENCE POSITION AGAINST SHUFFLED TEXT

Three sentence length range buckets (65-75, 75-85, 95-105) were omitted due to small sample size (n=6). Examples with less than 5 sentences each were omitted.

Table 5: ALiBi

| Sentence Length Range | Correlation | P-value | Number of Samples |
|---|---|---|---|
| 5-15 | -0.120560 | 1.037594e-24 | 904 |
| 15-25 | -0.083780 | 2.757708e-05 | 132 |
| 25-35 | -0.015695 | 5.596307e-01 | 48 |
| 35-45 | -0.037581 | 5.387906e-02 | 66 |
| 45-55 | -0.008077 | 4.455038e-01 | 178 |
| 55-65 | -0.019355 | 1.426657e-01 | 98 |

## D LINEAR REGRESSION SENTENCE RECONSTRUCTION BASELINE

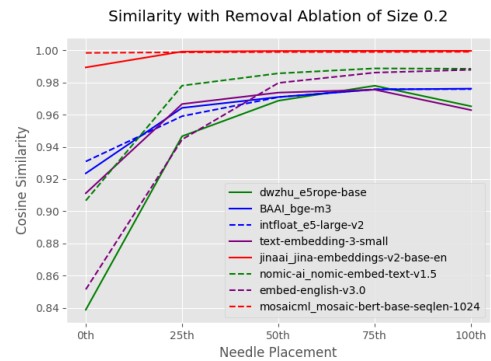
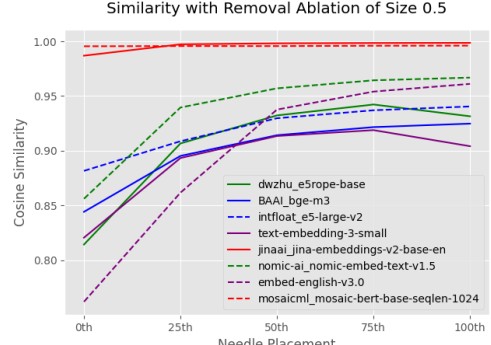

Table 6: APE

| Sentence Length Range | Correlation | P-value | Number of Samples |
|---|---|---|---|
| 5-15 | -0.204936 | 1.196681e-88 | 904 |
| 15-25 | -0.123513 | 1.420863e-18 | 132 |
| 25-35 | -0.036560 | 2.585037e-02 | 48 |
| 35-45 | -0.034370 | 7.942209e-04 | 66 |
| 45-55 | -0.009526 | 5.458451e-02 | 178 |
| 55-65 | -0.004620 | 4.229611e-01 | 98 |

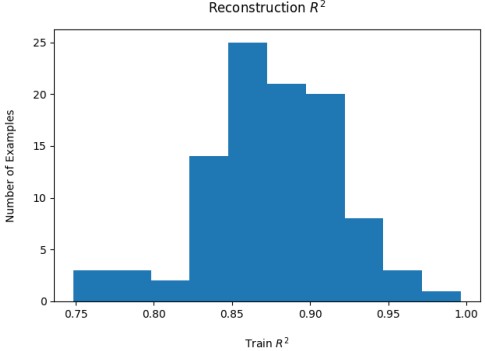

Table 7: RoPE

| Sentence Length Range | Correlation | P-value | Number of Samples |
|---|---|---|---|
| 5-15 | -0.201598 | 3.154022e-69 | 904 |
| 15-25 | -0.098903 | 1.669302e-11 | 132 |
| 25-35 | -0.044463 | 8.444829e-03 | 48 |
| 35-45 | -0.021359 | 4.203218e-02 | 66 |
| 45-55 | -0.009357 | 6.387572e-02 | 178 |
| 55-65 | -0.008881 | 1.290475e-01 | 98 |

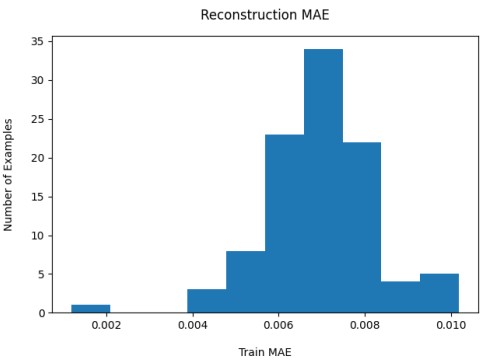

