# OpenReview forum: "Bias Learning: Quantifying and Mitigating Position Sensitivity in Text Embeddings"
_ICLR.cc/2025/Conference — Submitted to ICLR 2025_

### Official Review · Reviewer_qyKj · 2024-10-28

**Soundness:** 3
**Presentation:** 2
**Contribution:** 3
**Rating:** 6
**Confidence:** 4

**Summary:**

This paper investigates the impact of content position and input size on text embeddings and reveals current mainstream embedding models disproportionately prioritize the initial portion of the input text. Extensive ablation studies and regression analyses are conducted to investigate the position bias issue.

**Strengths:**

- This work investigates an interesting phenomenon: mainstream text embedding models disproportionately prioritize the initial portion of the input.
- Comprehensive ablation studies and regression analysis are conducted to research the bias position issue in text embeddings.
- The findings are helpful for text embeddings and suggest a future research direction for long-context embedding models.

**Weaknesses:**

- For text embeddings with CLS pooling, such as bge-m3, bge-large-en-v1.5, and UAE-Large-V1, prioritizing the initial part of the text should be advantageous. For text embeddings with average pooling, which part is prioritized doesn't seem to matter since it's global pooling.
- As claimed embeddings are important for information retrieval (IR) and semantic textual similarity (STS), but there is no experiment to show how such an initial priority phenomenon affects STS or IR tasks.

**Questions:**

### 1. Questions

- I am confused by this claim “Embedding models, in contrast, are theoretically position-invariant, due to their lack of the causal attention mask”. To the best of my knowledge, there are some LLM-based text embeddings like [1] with causal attention mask, although the causal attention mask may limit the performance of the embedding on STS tasks [2].
- I wonder if inserting irrelevant text or removing text will affect the coherence or change the semantics of the original text.
- Long-context embeddings tend to capture topic-level semantics. it is more coarse-grained than short-context. Perturbation may not show significant changes in the results. Since two of the Alibi models are longer-context (>512 tokens), that might be the reason why it is more robust than RoPE. Could you separate the reported performance between long-context and short-context models for Table 2?
- The selected embedding models are BERT-based and contain the special token CLS at the initial. Is it a possible reason why it prioritizes at the beginning? I guess the priority depends on the selected backbone. It would be appreciated if you could experiment with autoregressive LLM-based embeddings to see whether they will be prioritized in the last part of the text. This will make this work more comprehensive.


**Reference**

[1] Lee, J., Dai, Z., Ren, X., Chen, B., Cer, D., Cole, J. R., ... & Naim, I. (2024). Gecko: Versatile text embeddings distilled from large language models. arXiv preprint arXiv:2403.20327.

[2] Li, X., & Li, J. (2024, June). BeLLM: Backward Dependency Enhanced Large Language Model for Sentence Embeddings. In Proceedings of the 2024 Conference of the North American Chapter of the Association for Computational Linguistics: Human Language Technologies (Volume 1: Long Papers) (pp. 792-804).


### 2. Typo and Suggestions

- L20: with with -> with
- L39: “needles” -> ``needles’’
- Citation format: use \citep in L184, L191, L194, and L197.
- It is better to provide the full name of APE and RoPE in the abstract.
- In the experiment, it is better to take the pooling strategy into account.

---

### Official Review · Reviewer_B58k · 2024-11-04

**Soundness:** 2
**Presentation:** 3
**Contribution:** 2
**Rating:** 3
**Confidence:** 4

**Summary:**

This paper investigates positional sensitivity in encoder-based models. Through text insertion and removal experiments, the paper reveals that embeddings exhibit a bias towards early text positions, especially in models using APE and RoPE. It performs regression analysis to eliminate the effects of human writing styles. It proposes PADS method to help mitigate this bias.

**Strengths:**

1. Previous work mainly investigates positional bias in encode-decoder models while the work focus on positional bias in embedding models.
2. The paper is clearly written, providing a well-organized presentation of embedding models and positional encoding techniques.

**Weaknesses:**

1. Although the authors claim their regression analysis controls for human writing style, it’s not entirely convincing. Human writing conventions (e.g., important content at the beginning and end) are likely embedded in models trained on large corpora. Testing with shuffled sentences may not fully isolate this influence from model bias.
2. The results show ALiBi’s greater robustness compared to APE and RoPE, yet the paper does not investigate the underlying reasons, which would enhance its analytical contribution.

**Questions:**

1. Could the authors clarify how shuffled sentence embeddings effectively remove human writing biases?
2. Could they demonstrate PADS’s benefits on real-world IR tasks to show practical value?

---

### Official Review · Reviewer_E8un · 2024-11-04

**Soundness:** 3
**Presentation:** 3
**Contribution:** 3
**Rating:** 6
**Confidence:** 3

**Summary:**

This paper investigates positional biases in embedding models used in information retrieval (IR) and semantic similarity tasks, revealing that these models give disproportionate weight to the beginning of text inputs. By conducting experiments with insertion and deletion of irrelevant text at various document positions, the authors find that text at the beginning influences the model’s output embeddings more significantly than text in later sections. They attribute this bias to positional encoding techniques and training practices. To address this, the paper proposes a novel data augmentation method called Position-Aware Data Sampling (PADS), which mitigates the effect of positional bias, thereby enhancing model robustness for longer texts.

**Strengths:**

- The paper provides a comprehensive investigation of positional bias across multiple embedding models, input sizes, and document types.
- The Position-Aware Data Sampling (PADS) method is an innovative proposal to counteract positional bias, and the experiments show measurable improvements.
- The paper uses diverse datasets, making validation of the conclusions sonds.

**Weaknesses:**

- The analysis is limited to a few embedding models and positional encoding types, which may restrict generalizability to other architectures or languages.
- Cosine similarity as the primary evaluation metric may not fully capture how positional bias affects end-task performance (e.g., retrieval accuracy, relevance ranking).

**Questions:**

- Could PADS be integrated directly into model pre-training, rather than as a post-training adjustment? If so, how would this affect computational efficiency and model performance?
- How might the proposed approach impact the design of long-context models?

---

### Official Review · Reviewer_ArZK · 2024-11-05

**Soundness:** 1
**Presentation:** 3
**Contribution:** 1
**Rating:** 3
**Confidence:** 5

**Summary:**

This paper studies the phenomenon of positional bias in text embeddings. It observed that perturbations at the beginning of the texts affect the embeddings more than the changes at other parts of the texts. To reduce the positional discrepancy, a position-aware data sampling technique is proposed, where parts of the training data are sampled from later parts of the texts. Experiments show that after post-training with the technique, the resulted embeddings show reduced sensitivity to positions.

**Strengths:**

1. The positional bias in text embeddings can be an important aspect for research in long embedding models.
2. The paper is clear written and easy to read.

**Weaknesses:**

1. The experiments are lacking. The paper does not evaluate any text embedding benchmarks for retrieval performance. Many easy-to-run pipelines exist to evaluate on benchmarks such as MTEB, and there is no excuse to leave it to future work. It is important to evaluate on real retrieval task because it is unproven whether positional bias is harmful or not since such bias might naturally exists in real data.
2. For section 4.3, the writing bias exists in the training data. To argue that the cause of the position bias is not human writing, it is better to shuffle the training data, train another embedding model and to see if the resulted embeddings still exists in the newly trained model.

**Questions:**

N/A

---

### Meta-Review · Area_Chair_77iQ · 2024-12-19

**Metareview:**

The paper presents an observation that sentence embeddings are more susceptible to changes at the beginning of the text. Then, the authors propose a Position-Aware Data Sampling technique to mitigate the bias.

The main drawback, as mentioned by reviewers, is that the paper only did cosine similarity analysis, instead of showing the effectiveness in downstream tasks (such as retrieval).

**Additional Comments On Reviewer Discussion:**

Reviewers gave either a borderline score or a rejection score. Authors did not provide a rebuttal.

From my point of view, not showing downstream performance is a major drawback, so I am recommending rejection.

---

### Decision · Program_Chairs · 2025-01-22

Reject